# Engineering and Geophysical Research of the Tailing Dump under the Conditions of Growing Soils of the Base

Kristina Tulisova [1,*], Vladimir Olenchenko [1], Nikolay Sigachev [1,2], Nikolay Yurkevich [1], Nataliya Yurkevich [1] and Tatyana Kuleshova [1]

[1]  Trofimuk Institute of Petroleum Geology and Geophysics SB RAS, Novosibirsk 630090, Russia
[2]  Faculty of Construction and Ecology, Transbaikal State University, Chita 672039, Russia
*   Correspondence: tulisovaky@ipgg.sbras.ru

**Featured Application: Assessment of environmental risks in connection with the technogenic thawing of permafrost soils in the places of waste storage of mining enterprises in the Far North.**

**Abstract:** The relevance of the work is due to the risks of an uncontrolled increase in circulating water leaks through sides and bed of the dam, caused by thawing of permafrost soils in the Far North. The main aim of the work is to scientifically substantiate a set of engineering measures to reduce filtration consumption and restore and maintain the waterproofing of the tailing dump. The object of the study was the tailing dump of the concentration plant, with adjoining filter walls. The tailing dump has been exploited since 1996; for the last 20 years, circulating water leaks into the shunting tank located below were recorded. Within the water area of the tailing dump and at the landfalls, geophysical surveys were carried out from ice by the TEM (transient electromagnetic) method. The obtained geoelectric sections made it possible to form a holistic view of the structure of the filtration zones in the right and left bank junctions. The data obtained will be used for planning anti-filtration arrangement.

**Keywords:** tailing dump; permafrost; soil thawing; filtration; transient electromagnetic method

## 1. Introduction

Wastes from mining and processing enterprises become sources of environmental pollution, including due to the oxidation of finely divided tailings of ore enrichment with air and water oxygen with the formation of mineralized drainage waters [1–4]. In addition, the processes of interaction "water-tails" in the conditions of the Far North often lead to the thawing of permafrost soils, the formation of technogenic taliks, and the leakage of technogenic waters, which, in the presence of fault zones, leads to the development of filtration channels [5].

During the operation of the processing plant of the diamond mining enterprise since 1996, the permafrost soils of the landfalls both on the left and on the right side of the tailing dump have thawed, through which, starting from the period 2000–2001, leakage of significant volumes of circulating water into the lower shunting tank was recorded. The development of filtration channels in the fractured tectonic zones that make up the near-edge mountain range does not stop [1–4].

Prevention of a dangerous development of the situation at the tailing dump requires additional research, as well as the development of technological solutions for the maximum possible elimination of filtration channels [5–8].

The solution of these problems will not only ensure the environmental safety of the tailings operation, but also reduce the costs of its operation, which are largely determined by the energy costs for the return of circulating water leaks from the shunting to the main tailings storage.

Geophysical methods allow solving a wide range of problems. In particular, in geoecology, geophysical methods are used to optimize the monitoring system of technogenic systems by reducing the number of samples taken for geochemical studies, assessing the resources of useful components, and delineating the distribution zones of underground drainage flows [9,10].

To control the state of tailings and to prevent emergency situations, it is advisable to use geophysical research methods. Since Soviet times, geophysical methods were widely used in the construction and operation of hydraulic structures. As for regime geophysical observations, they are not carried out on the vast majority of hydraulic structures located in the northern and northeastern regions of Russia, where permafrost soils are widespread (≈65% of the total territory of the country). An exception to the general rule is objects that are in the sphere of influence of PJSC (public joint-stock company) Alrosa and some foreign companies [11,12].

This study explored the determination of filtration zones in soil dams by electromagnetic sounding method(TEM method). Main advantages of the TEM method—the absence of galvanic grounding, the possibility of working in winter, the locality of research, the absence of a shielding effect from the upper high-resistance frozen horizon [13,14].

The criterion for distinguishing thawed rocks against the background of frozen ones is their low electrical resistivity. The resistivity of thawed rocks depends on the lithological composition, humidity and mineralization of pore moisture. The same parameters determine the resistivity of frozen rocks, but their temperature is the main factor [12]. In the study area, thawed technogenic soils (tailings) are widespread, which are represented by medium- and fine-grained sands, gravel, as well as alluvial sandy loams and weakly permeable, saline clays. The base of the tailings and landfalls are composed of carbonates—limestones and marls, the resistivity of which will depend on the degree of their fracturing, frozen or thawed state.

According to the results of many years of research using the method of electrical tomography, within the boundary dam and the left bank junction, the resistivity of rocks varies in a wide range—from 10 to 10,000 Ohm·m [15]. The lowest resistivity values are thawed areas of bulk, bedrock deposits confined to a natural talik formed due to the influence of water basins of the hydrotechnical system. Frozen bedrocks are characterized by increased resistivity values, as well as bulk coarse-grained soils in the upper part of the section.

The primary task of geophysical research was to identify in plan and trace to depth the fault zones through which water leaks from the reservoir. The information obtained is used to clarify the boundaries of fault zones, their types of fracturing, and filtration characteristics. Taking into account these data, it is possible to plan certain impervious measures.

## 2. Materials and Methods

### 2.1. Case Study

The studied case was located on the territory of Siberia in the permafrost zone (Figure 1). The case was the tailing of the processing plant, where waste is stored after diamond mining. The work site was located in the area where the protective dam and the left side of the tailing dump adjoined.

According to engineering and geological surveys carried out in 1985–1986 [16], the geological structure of the territory includes deposits of the Upper and Middle Carboniferous of the Kat Formation ($C_{2+3}$ kt); the Landoverian of the Lower Silurian ($S_1$ ln); Ordovician deposits of the lower and middle sections of the Sokhsoolokh ($O_1$ ss) and Kylakh ($O_2$ kl) suites, overlain by a cover of Quaternary deposits of lacustrine marsh, eluvial-deluvial, alluvial and eluvial genesis.

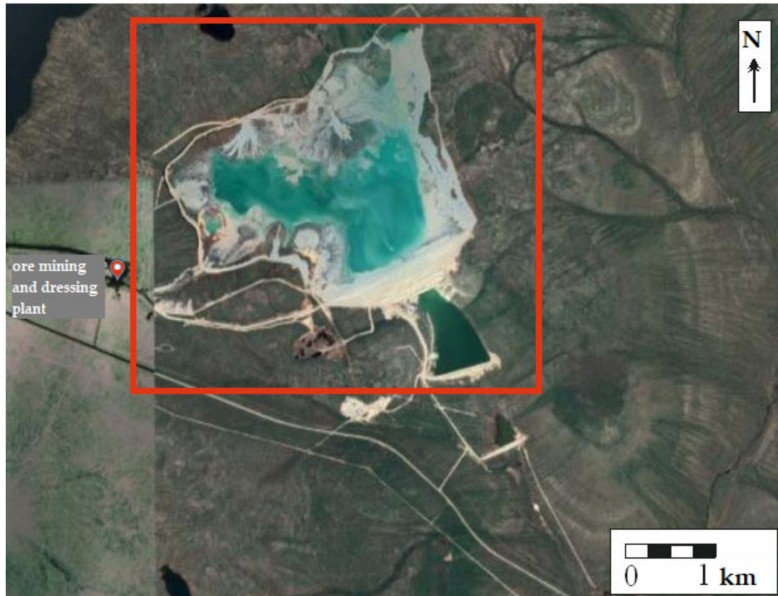

**Figure 1.** Satellite image of the study area (the tailings dump is highlighted in a red frame).

Geomorphologically, the territory of the tailing dump is located in the trough-shaped valley of the Fossil Creek. The absolute elevations of the surface vary on the starboard side from 602.24 to 503.05 m. On the left side, they vary from 609.52 to 503.05 m. The elevation difference was 99.19 and 106.47 m, respectively [16].

In geocryological terms, the study area was located in the area of distribution of permafrost of the merging type with rock temperatures from $-2.6$ to $-6.6$ °C and a thickness of 200 to 600 m. At the time of surveys in 1985, the presence of a buried ice and ice-rich soils. Based on the results of the survey, physical and geological processes and phenomena unfavorable for construction were established, such as kurum, frost heaving, heavily icy soils and buried ice, waterlogging, and tectonic disturbances [16].

Within the site of the enclosing dam, 6 zones of discontinuous faults of various thicknesses were identified. The strike of the zones was 150–160°, the dip was east at an angle of 60–70°.

The tailing dump is a hydraulic engineering structure of the I class of solidity, according to the location—beam, according to the method of filling—alluvial. The tailings storage capacity at the final level of 605.0 was 230.2 million m$^3$.

The tailing dump of the processing plant was put into operation in 1996. This concentrating plant processes ore mined from the diamond pipe. The design capacity of plant was 10 million tons of ore per year.

Tailing dump design parameters:

- The design length of the enclosing dam at the final filling level was up to 2500 m (1900 m in 2018);
- The design height was 105 m (97 m in 2018). The protecting dam was thawed alluvial. The tailing dump was drainless, full-circulation, with the accumulation of circulating water for the entire period of operation.

Studies within the sites of the planned construction of the enclosing and pioneer dams in the right and left sides of the valley of the Fossil Creek revealed zones of tectonic disturbances (crushing). Strike zones were 150–160°, dip 60–75°. The thickness of the zones reached 160–180 m. Figure 1 shows the results of the interpretation of the observed processes of suffusion and zones of tectonic faults, identified as a result of previous geophysical studies and the engineering-geological survey of 1986, carried out by specialists from the mine surveying, geological and hydrogeological services.

Analysis of engineering-geological surveys, schemes of tectonic disturbances (Figure 2), allowed us to draw the following conclusions:

1.  There are fault systems in the base rocks;
2.  There are several fault crossing nodes in the left bank junction;
3.  When the fixed suffosion processes and tectonic disturbances are superimposed, a fairly clear correlation can be traced;
4.  Downstream on the port side, groundwater is currently being discharged;
5.  The unloading area is located at the intersection of the general and feathering fault zones.

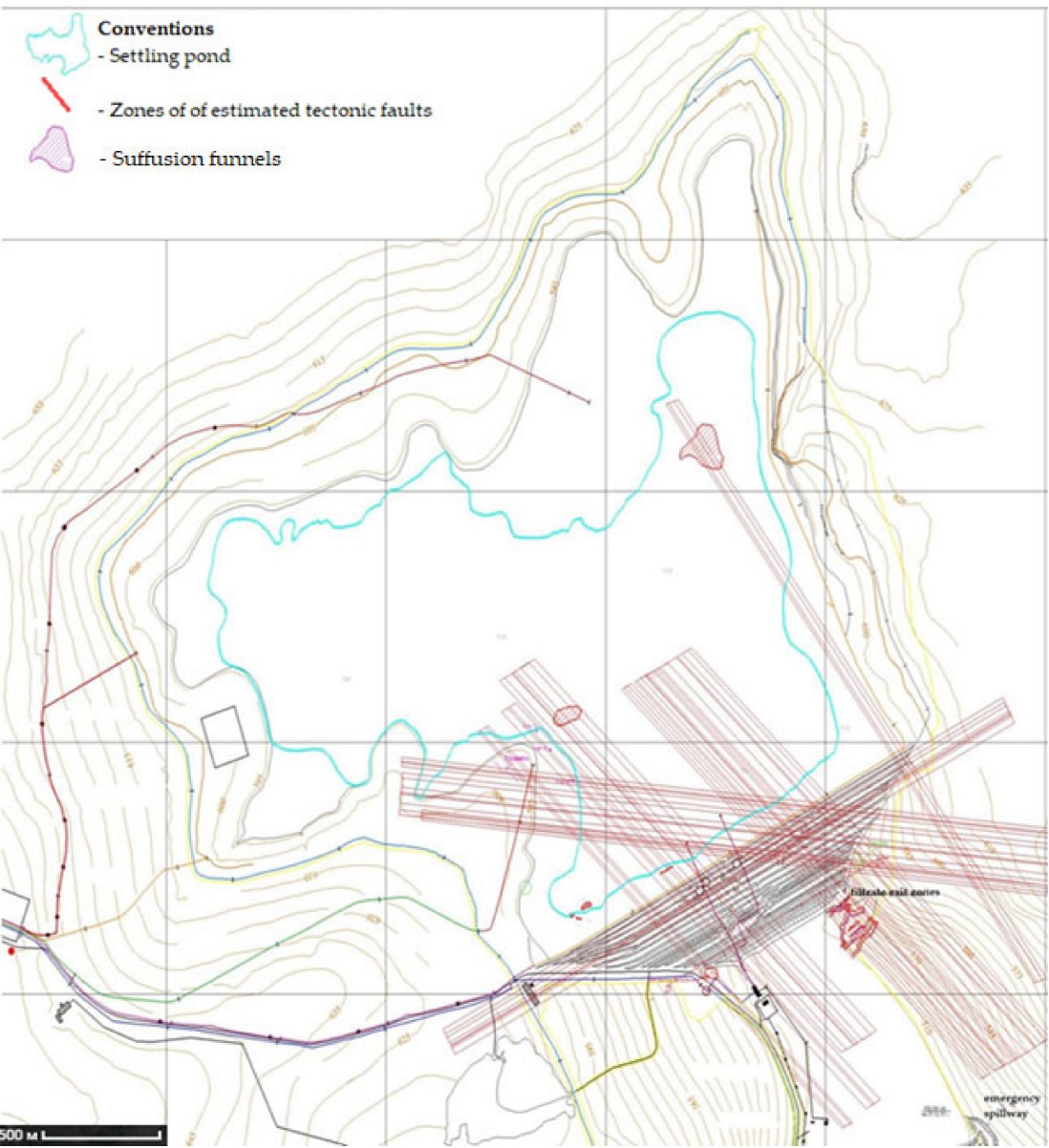

**Figure 2.** Scheme of estimated tectonic faults (colored thick lines—communications of the processing plant with the tailings).

Separately, it should be noted that, according to the results of drilling, the crushing zones are represented by rubble-grus fragments of carbonate rocks (most often limestones), with the addition of terrigenous material (subaerial loams). In the undisturbed state, the fragments are cemented by wedge ice, and the ice content of the formed deposits is up to 50% (Figure 3).

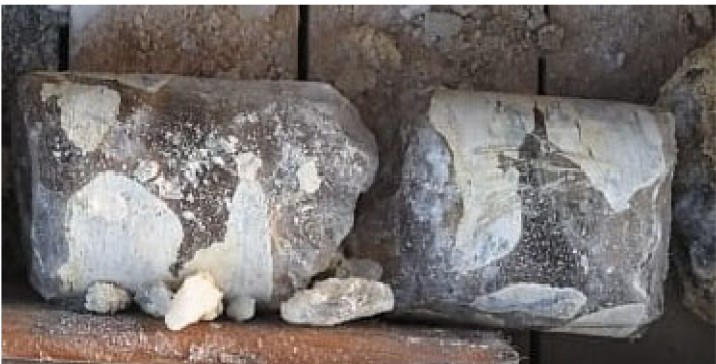

**Figure 3.** Ice-cemented carbonate breccia (core sample from the right abutment of the tailing dump).

The high ice content is because before undergoing significant changes associated with the cryolithogenesis of this area in the Quaternary period, the studied interval belonged to the zone of intensive water exchange of the first fresh aquifer from the surface [16].

Three genetic types of cracks were identified in the rock mass: lithogenetic, exogenous, and tectonic (Figure 4).

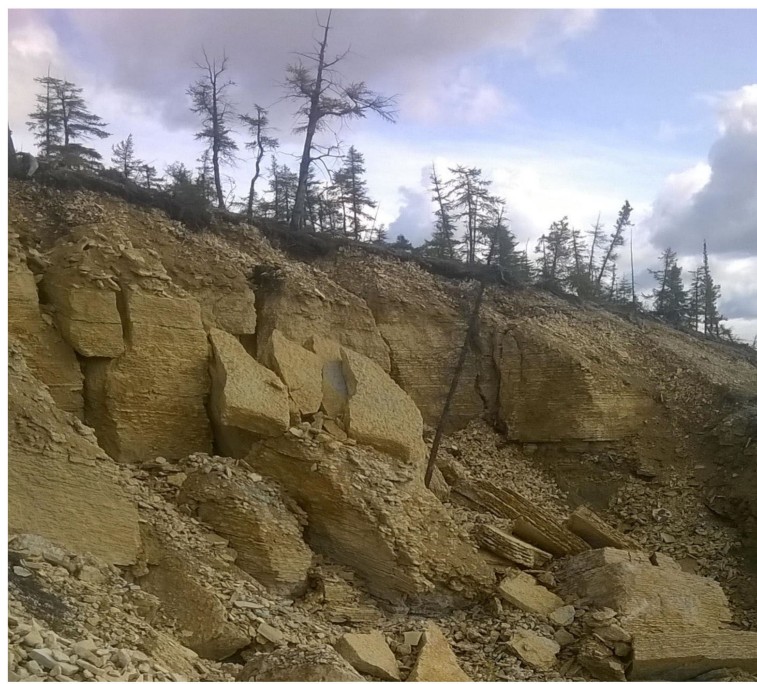

**Figure 4.** Silurian limestones broken by fracture systems.

Subhorizontal lithogenetic cracks were traced to a depth of 10 m. They were filled with ice and their width was 1–3 cm. When soils with exogenous fractures thaw, the rocks crumble into rubble. Tectonic cracks in fault zones have a chaotic arrangement. Rocky soils are broken along these cracks into acute-angled fragments. Crack filler—ice or loam.

According to the symmetrical electrical profiling data, the linear zones of low apparent electrical resistivity are tectonic faults, confirmed by drilling [16].

*2.2. Methods*

Time-domain electromagnetic (TEM) sounding is a method of electromagnetic research based on the study of the field of transient processes, which is excited in the earth during a pulsed switching of the current in the source [14].

Probing by the formation of the field refers to methods with an artificial (controlled) source. The sources can be a horizontal electric dipole (grounded electric line AB) or a vertical magnetic dipole (ungrounded current loop Q). Either a grounded electrical line (MN) or an ungrounded loop (q) is also used as a receiver. In our case, loops were used for the source and destination.

To excite the field of transients, a pulsed current switching was created in the supply (generator) loop. With an instantaneous shutdown of the current in the supply loop, the voltage measured in the receiving loop did not drop to zero instantly, but gradually, changing in a rather complex way.

This is explained by the fact that at the moment the current was turned off, secondary currents were induced in the conductive regions of the cut. The alternating magnetic field of the secondary currents induced an electromotive force (EMF) in the receiving loop. Moreover, the EMF in the receiving loop was proportional to the rate of change of the magnetic flux. At the initial moment of time (at short measurement times after switching off the current in the supply loop), secondary currents were distributed in the near-surface part of the section. With the passage of time (in a pause after turning off the current in the supply loop), the currents began to penetrate into deeper layers, fading away from the source. This process is called the formation of the field in the earth, and the dependence of the voltage in the measuring loop on the time elapsed since the current switching in the supply loop is called a transient response.

Thus, the depth of penetration of the transient field into the ground is determined by the time elapsed since the current was turned off in the generator loop, which is called the delay time or the rise time. This property makes it possible to carry out soundings by studying the dependence of the components of the measured electromagnetic field on the delay time.

At the research site, measurements of the characteristics of transient processes were performed using a Fast-Snap digital telemetric electrical survey station [17]. The station was intended for the registration of electromagnetic signals (transients) during work by the TEM method. It includes the following blocks: an on-board module as part of a notebook class computer and a communication line adapter; current synchronization and measurement device (current switch); three telemetric meters (receiver) with measuring loops.

During probing, the size of the generator loop was $100 \times 100$ m. The electromagnetic field was recorded by three induction sensors—$1 \times 1$ m in size, equivalent in magnetic moment to a single-turn loop $50 \times 50$ m in size. The sensors were located inside the generator loop at a distance of 50 m from each other. The measurement scheme for profiles is shown in Figure 5. The installation was moved manually.

When performing work, the generator loops were located so that the diagonal coincided with the direction of the profile. The measurements were carried out simultaneously by three channels: one coaxial sensor and two offset by 50 m in the direction of the setup. The step of the generator loop along the profile was 150 m. The total amount of work carried out by the TEM method was 605 basic physical observations and 31 control physical observations.

Primary data processing was carried out in specialized software for Fast-Snap equipment [18]. The quantitative interpretation of the data was performed using the TEM-IP program developed at the IPGG SB RAS [19].

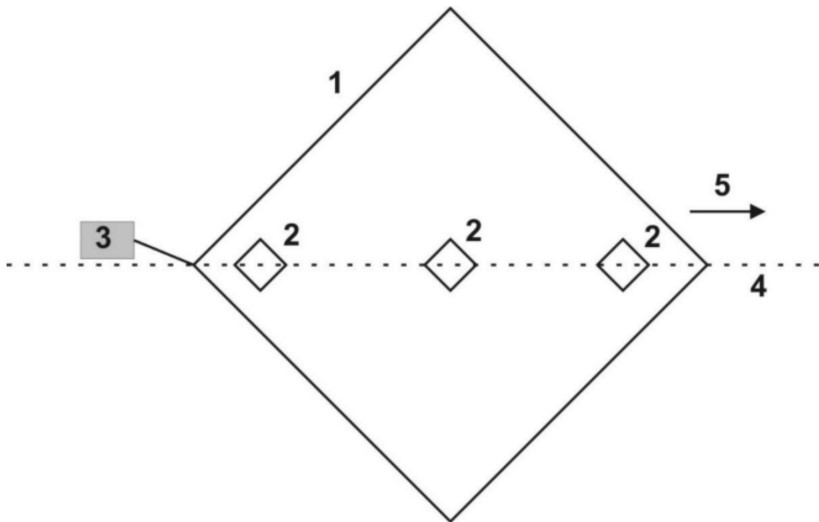

**Figure 5.** Scheme of measurements by the TEM method along the profile. 1—generator loop, 2—induction sensors with meters, 3—current switch and laptop, 4—profile, 5—direction of movement along the profile, dotted line—profile line.

An example of experimental and theoretically fitted curves within a one-dimensional horizontal layered model was generated (Figure 6a). If the curve was complicated by fast-decaying induced polarization (IP), the solution of the inverse problem was carried out taking into account the IP (Figure 6b).

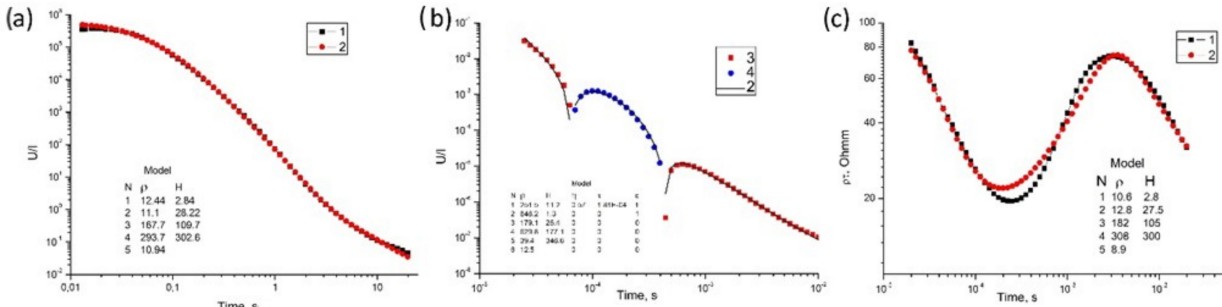

**Figure 6.** Examples of the curve for a one-dimensional horizontally layered medium at point 330 (**a**), RMS 4.6 %; the curve complicated by the effect of IP in the form of a signal transition to a negative area at point 340 (**b**) RMS 9.3 %; apparent resistivity curve complicated by the three-dimensional inhomogeneity of the medium at the picket 334 (**c**), RMS 12%. (1 is the experimental curve; 2 is the theoretical curve; 3 is the positive signal; 4 is the negative signal. Model: $\varrho$ is the resistivity of the layer (Ohm·m); H is the layer thickness (m), $\eta$ is the chargeability ($0 \leq \eta \leq 1$), $\tau$ is the relaxation time (ms), c is the exponent ($0 < c \leq 1$)).

Some transient responses were poorly fitted to the theoretical curve, even taking IP into account. Such cases were noted at points close to the coastline. There were inclined or vertical boundaries of thawed, and frozen rocks were located close to the points here. Therefore, the influence of a steeply dipping interface between media with different resistivity was expressed as a distortion of the right ascending branch of the curve, which increased at a very steep angle. Figure 6c shows an example of an apparent resistivity ($\varrho$k) curve complicated by a three-dimensional inhomogeneity of the medium. On the experimental curve, the right ascending branch increased so rapidly with time that such a curve cannot be described by a theoretical curve in the framework of a one-dimensional horizontally layered model. The theoretical curve closest to it with standard deviation corresponded to the five-layer model.

The primary task of geophysical research was to identify in plan and trace to depth the fault zones through which water leaked from the reservoir. After receiving this information, drilling was carried out to clarify the boundaries of fault zones, types of their fracturing, and filtration characteristics. Taking into account these data, it is possible to plan certain impervious measures.

According to the resistivity measured at direct current, two stratigraphic-genetic complexes were distinguished (Table 1).

**Table 1.** Specific electrical resistivity of rocks within the study area according to [15].

| Rock Complex | Resistivity, Ohm·m | |
|---|---|---|
| | **Thawed** | **Frozen** |
| Technogenic bulk and alluvial soils | 10–300 | 1000 |
| Limestone dense fissured, weathered marl | 50–2200 | 2400–10,000 |

As a rule, the resistivity of rocks, determined by the TEM method, was somewhat lower than the resistivity measured at quasi-direct current. In addition, the TEM method was weakly sensitive to changes in resistivity in high-resistance media, where $\rho$ exceeds 500 Ohm·m.

According to the experience of our studies using the TEM method, 40 km east of the Aikhal settlement [20], frozen carbonates had resistivity from 500–800 to 1100–1200 Ohm·m. In areas of permafrost degradation, their resistance dropped to 37–95 Ohm·m. Below the permafrost base, the resistivity of rocks decreased to 0.5–18 Ohm·m, and in the brine horizons—to 0.5–0.8 Ohm·m.

The results of soundings by the other researchers TEM method [15] on the left bank junction showed that frozen carbonates had a resistivity of 300–1000 and more Ohm·m. At a depth of about 180 m, below the permafrost base, the resistivity of rocks varied within 5–25 Ohm·m.

Thus, within the study area, frozen and thawed rocks differed well in resistivity. For the TEM method, the electrical resistivity of frozen bedrock will lie in the range of 300–1500 Ohm·m, depending on the lithological composition, temperature, and salinity. The resistivity of thawed carbonates will vary from 35–100 to 250 Ohm·m, depending on the fracturing and water cut. The resistivity of thawed technogenic soils will have values within a few tens of Ohm·m, depending on the particle size distribution and humidity.

As an example of applying the interpretation criteria, a geoelectric section along profile No. 1 was presented (Figure 7). The interpretation of geoelectric sections was carried out up to an absolute elevation 300 m since the deeper rocks were resistivity homogeneous.

In the central part of the sections, in the depth interval of 0–50 m, we observed low-resistivity technogenic deposits with $\varrho$ 9–18 Ohm·m. The high-resistivity layer (300–1000 Ohm·m) was interpreted as permafrost. Its thickness was 200 m on the western coast and 250 m on the eastern connection. We did not observe the boundary between bedrock complexes of different lithologies on the section—limestones and marls, since in the frozen state their resistivity was equally high from the point of view of the resolution of the TEM. The high-resistivity layer was broken up by steeply dipping narrow low-resistivity zones, which were interpreted as crushing zones through which groundwater was filtered. At the base of the section, from a depth of 200–250 m, a layer of very low-resistivity was identified—8–45 Ohm·m, which was interpreted as a saline stratum.

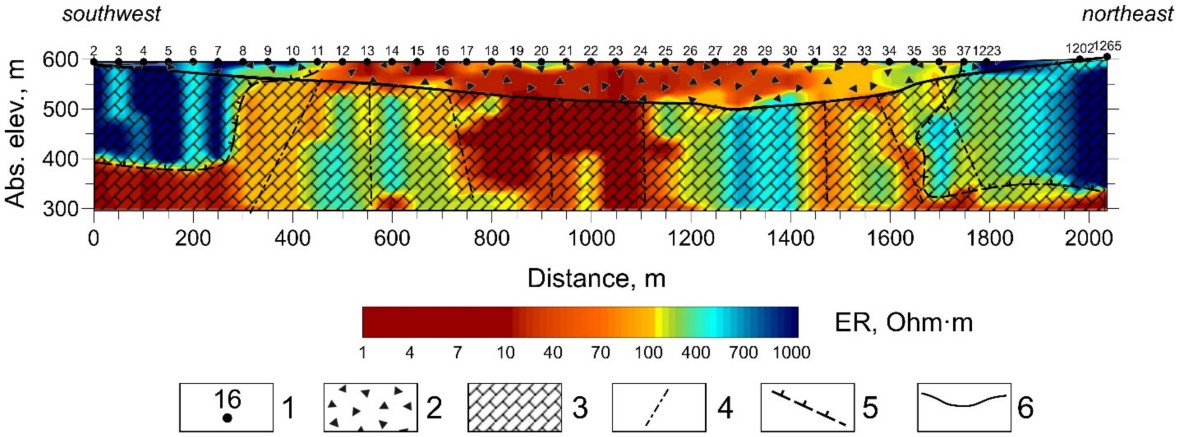

**Figure 7.** Geoelectric section along profile No. 1 with interpretation elements (1—item TEM and its number; 2—technogenic deposits (tailings); 3—carbonate rocks; 4—fault axes according to TEM data; 5—permafrost boundary according to TEM data; 6—relief surface before the start of filling the tailings).

## 3. Results

### 3.1. Interpretation of Resistivity Distribution Maps

Based on the results of 1-D inversion of the data of areal electromagnetic soundings, maps of the distribution of electrical resistivity of rocks at depths of 20, 30, 50, 65, 75, 100, 125 and 150 m were built.

On the resistivity distribution map at a depth of 20 m, technogenic deposits were distinguished by a vast area with a resistance of 9–15 Ohm·m in the central part of the study area. In the left-bank junction along the coast, a linear zone of low resistivity with a width of 70–130 m appeared. In the area of the enclosing dam, the linear anomaly merged with the areal anomaly of low resistivity from technogenic flooded soils. The linear anomaly of low resistivity was interpreted as a zone of fractures, along which bypass filtration of water occurred in the left bank junction. It was clearly seen that the zone of formation of cracks on the left bank fell on the boundary between low and high resistivity rocks. On the right-bank junction, a linear anomaly of low resistivity appeared fragmentarily along the coastline, along which bypass filtration was likely to develop. In 2019, there was an outflow of water in the downstream on the starboard side at 533 and 531 m.

At a depth of 30 m, linear filtration zones in the landfalls were more pronounced. A depth of 30 m corresponded to a height of 565 m, at which a crack formed in the downstream at the junction of the dam to the left bank. In addition to the pronounced linear zones of low resistivity, local anomalies of low resistivity (less than 100 Ohm·m) against the background of reduced electrical resistivity of rocks (250–300 Ohm·m) stood out in the study area in the tailings contour. The existence of local anomalies was explained by uneven thawing of icy soils at the base of the tailings.

At a depth of 50 m, the linear zone of no resistivity in the left bank junction shifted to the northeast, which was caused by its northeast dip. On the right bank of the tailing dump, the linear anomaly of low resistivity remained within the boundaries of the selected contour, which meant its vertical drop. In the central part of the area, there was an area of low resistivity associated with technogenic deposits. It was clearly seen that the rocks along the enclosing dam also had a low resistivity (20–50 Ohm·m), which was explained by their water saturation below the level of the depression curve.

Figure 8 shows the resistivity map at a depth of 65 m. This depth corresponded to a height mark of 530 m, at which a crack formed in the downstream on the left side and groundwater was discharged. Water outcrops were recorded at 520–526 m. At 530 m along the port side, there was a linear zone of low resistivity, probably associated with bypass filtration along fractured rocks. On the right-bank junction at the level of 530 m, local low-resistance anomalies were distinguished, which conditionally formed a linear

zone. On resistivity maps at depths of 75–150 m, the pronounced linear character of low-resistance anomalies in landfalls gradually disappeared. In the central part of the study area, under the tailings deposits, a low resistivity area was identified, probably associated with a deep talik at the base of the hydraulic structure. The distribution of resistivity over the area acquired a mosaic character, which can be associated both with uneven thawing at the base of the tailing dump, and with interpretation errors, when the influence of the three-dimensionality of the medium was expressed in the appearance of a false underlying layer of high resistivity in a one-dimensional model.

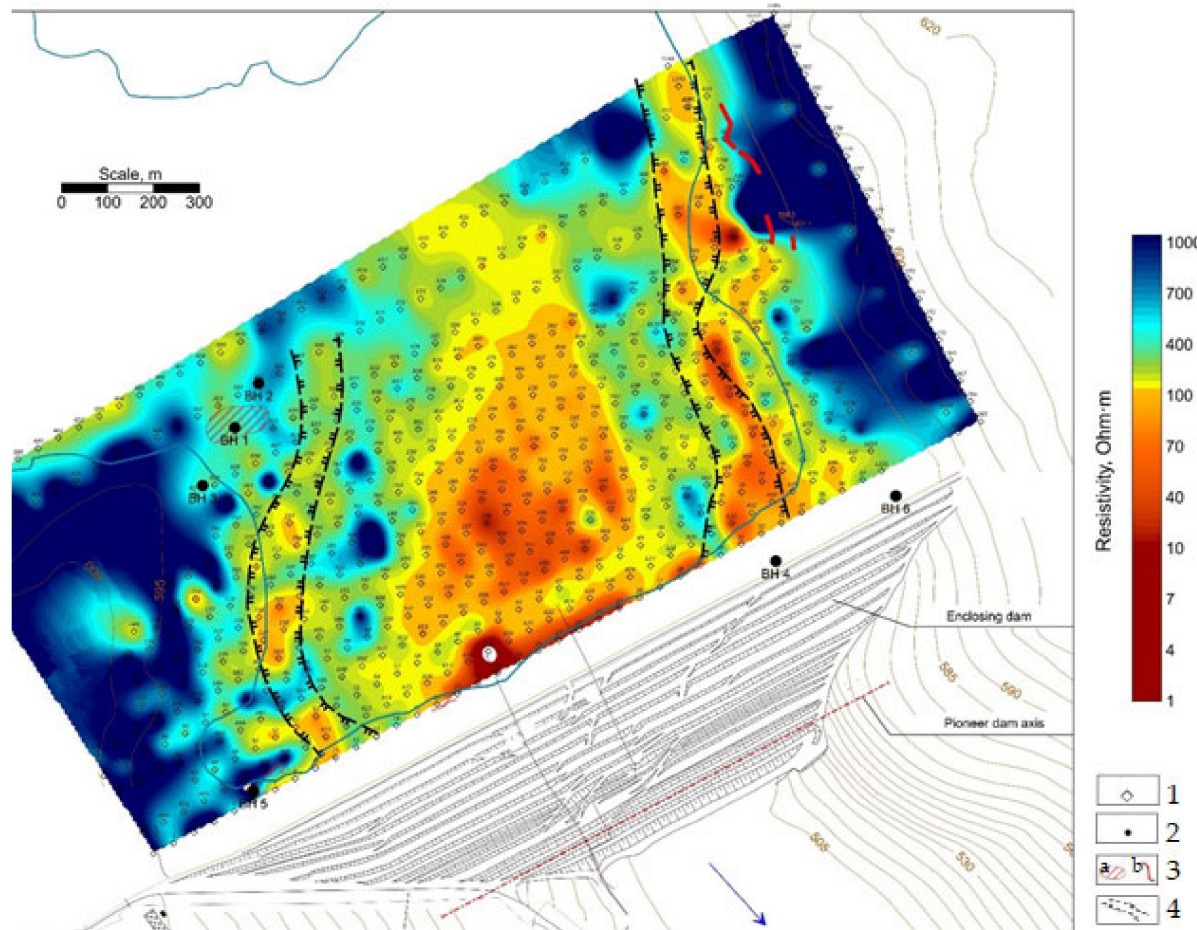

**Figure 8.** Resistivity distribution map at a depth of 65 m (absolute elevation 530 m). 1—points of the TEM; 2—wells; 3—dips (a) and cracks (b); 4—linear filtration zones according to the TEM data; the blue arrow shows the direction of the filtrate outlet.

On the resistivity map at a depth of 150 m in the zone of formation of a sinkhole in the right-bank junction, an anomaly of low resistivity was identified, associated with the subpermafrost complex of saline rocks.

Directly in the landfalls along the contour of the 605 m elevation, the resistivity of the rocks took on very high values, which was caused by their frozen state.

It should be noted that the places of formation of suffusion funnels (dips) along the right bank weakly correlated with resistivity anomalies. There was no clear relationship between the location of the formation of the failure and the resistivity of the surrounding rocks. The largest dip was located between profiles 15 and 16. Well No. 1 was drilled in it, which, to a depth of 17.5 m, uncovered low-moisture man-made soils, limestone crushed stone from a depth of 17.5 to 20.8, and massive dense frozen limestone. Below the section to a depth of 76 m, the rocks were frozen. That is, the formation of a dip here was not

associated with intense filtration along the fault. At the same time, the hydraulic connection between the sinkhole formation site and the outlet in the downstream on the starboard side was unequivocally established, since tailings wash-out by end discharged directly into the zone of filtration flows inlet (dip) led to a decrease in the filtration volume to 50–100 m$^3$/h in the lower pool [15].

According to the results of surveys in 1985, it is known that the eluvium of carbonates in the form of crushed stone was cemented by ice. Probably, the sinkhole on the modern surface was formed as a result of thawing of ice-cement in the weathering crust of carbonates, and water filtration initially occurs not along the fracture zone in bedrock, but along the thawed weathering crust. Where the talik develops along a fracture zone in carbonates, water enters the fault. Such narrow zones in the right-bank junction were clearly distinguished on geoelectric sections along profiles 2, 6–8.

The mechanism of formation of dips at the enclosing dam in the right-bank junction was also not obvious from the analysis of the structure of the section along well No. ice. From a depth of 27.7 m, the carbonates are frozen, with ice along cracks. Consequently, the formation of sinkholes here was not associated with water filtration along the fault, but was most likely caused by the melting of massive syngenetic ice in technogenic deposits, which was formed on a shallow beach in the process of filling the tailings.

The formation of some funnels along the left bank junction was accompanied by an increase in the debit of groundwater outflow in the downstream [15]. Here, the connection between the melting of ice-cement in the eluvium of carbonates and in the cracks of bedrock along faults was obvious. On 27 June 2018, on the left side of the tailing dump, 700 m from the left bank junction of the enclosing dam the reclaimed beach was deformed 50 by 100 m in size with the formation of cracks and subsidence. Two funnels formed, one directly in the beach area about a meter in size, the other in a pond about 5–7 m from the water's edge (Figure 9). The filtration volume in the landfall of the left slope of the shunting tank increased to 15,000 m$^3$/s. Backfilling with overburden rocks (limestone, marl) ~7000 m$^3$ of the visual entrance of the filtrate (funnel) in the pond zone of the tailing dump was promptly carried out. At the same time, a concentrated alluvium was organized with pulp outlets into the zone of the deformed beach. These measures managed to stabilize the seepage flow rate up to 3500 m$^3$/h within 5 days.

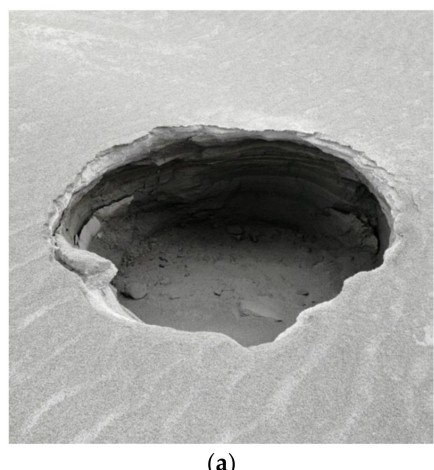
(**a**)
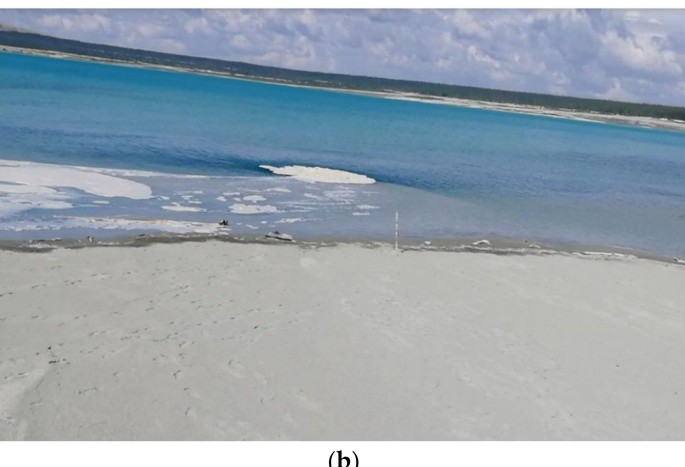
(**b**)

**Figure 9.** The sinkhole on the beach (**a**) and the place of the leachate entrance in the pond (**b**) on the left bank of the tailing dump [15].

The linear filtration zone was well distinguished in the left-bank junction on the resistivity distribution maps at depths of 20 and 30 m. Along this zone, the beach was deformed and craters were formed in 2018.

### 3.2. Volumetric Geoelectrical Model of the Site

The 3D geoelectric model of the study area was a set of one-dimensional models interpolated in three-dimensional space. Figure 10 shows the three-dimensional distribution of resistivity in the volume of the medium. The horizontal section of the model was made at an absolute elevation of 530 m. At this elevation, groundwater was discharged in the downstream at the right and left bank junctions.

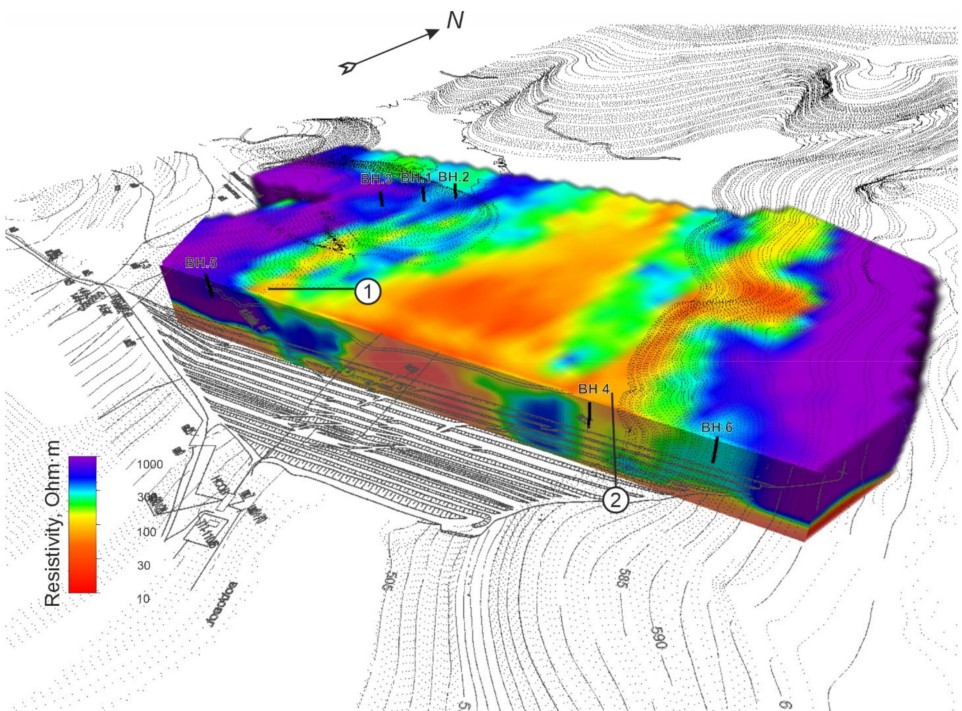

**Figure 10.** Volumetric geoelectric model of the research site: 1—filtration along the fault in the right-bank (western) junction; 2—the same in the left-bank (eastern) junction.

The overlay of the identified low-resistance zones on the relief elements of the valley of the Fossil Creek is shown in Figure 10.

As can be seen from Figure 10, a linear zone of low resistivity was traced in the left bank junction, along which water filtration and discharge in the downstream are expected. In the right-bank junction, the linear zone of reduced resistivity was less pronounced.

In the central part of the site, the area of low resistivity corresponded to technogenic deposits. At the same time, their resistivity varied from 30 to 150 Ohm·m. It was assumed that such a range of resistivity changes was associated with different moisture content of the tailings. An extensive anomaly of low (10 Ohm·m and less) resistivity in the section along the enclosing dam was associated with both water filtration and the influence of siphon water intakes. Thus, the three-dimensional geoelectric model made it possible to form a holistic view of the structure of the bypass filtration zones in the right and left bank junctions. Figure 11 shows a diagram of the location of faults and filtration zones, as well as the choice of places for injection of plugging compounds, compiled on the basis of the engineering and geophysical studies.

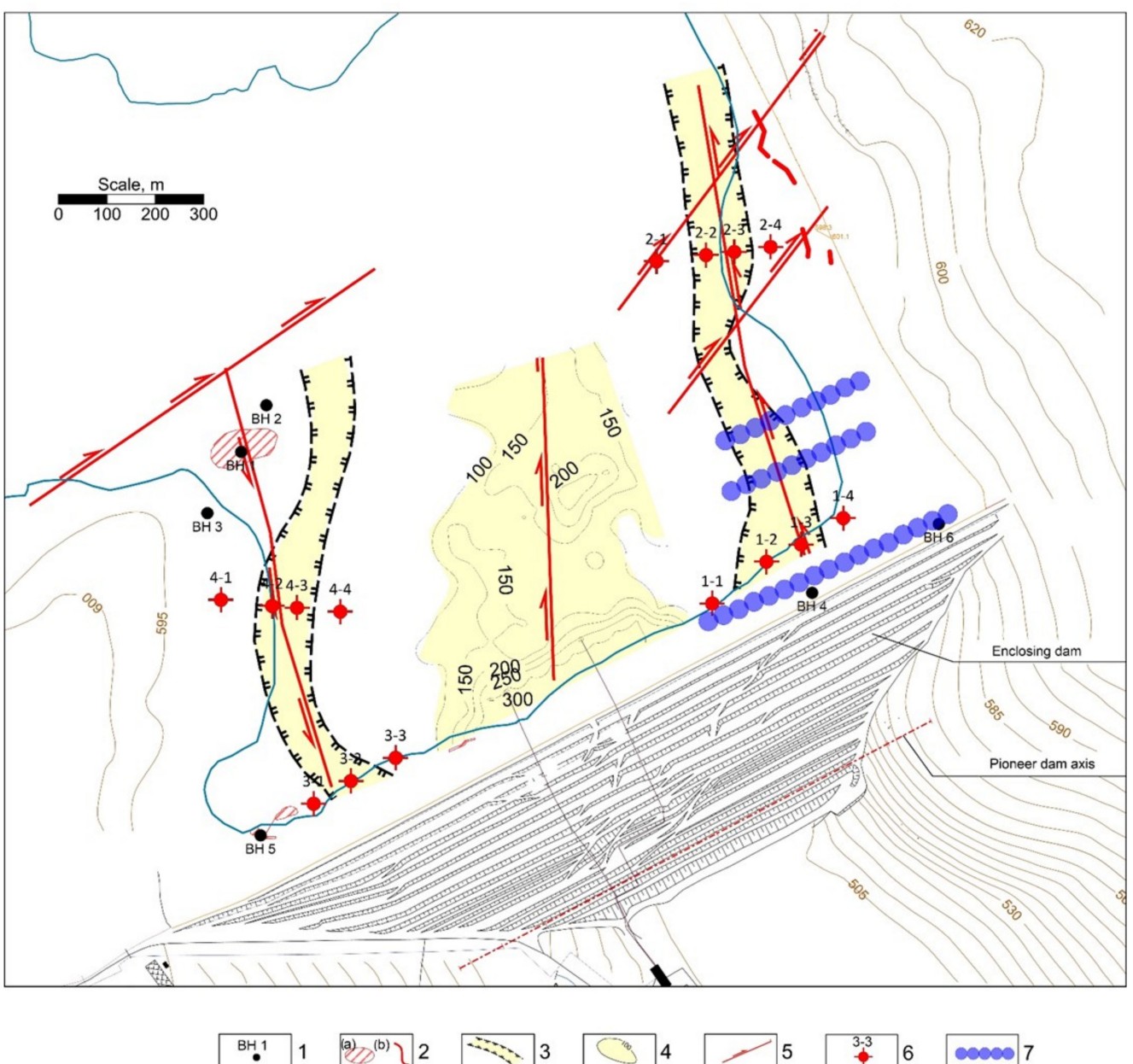

**Figure 11.** Layout of faults and filtration zones, as wells for injection of plugging composite. 1—wells; 2—dips (a) and cracks (b); 3—filtration zones in landfalls; 4—filtration zone at the base of the tailing dump and thawing depth; 5—faults; 6—recommended control wells; 7—wells for injection of plugging composite; blue line—contour of the tailings dump.

## 4. Discussion and Conclusions

On the area of the tailings of the enrichment plant, areal soundings were carried out using the TEM method in order to obtain information about the engineering and geological conditions of the territory, sufficient to assess and predict changes in these conditions. The objectives of the research were to determine the position of fault zones under technogenic deposits and in lateral junctions, to identify the proposed filtration zones.

As a result of the analysis of resistivity distribution maps, the following was established:

1. In the left bank junction at depths of 20–50 m, a linear anomaly of low resistivity was revealed, which was interpreted as a zone of bypass filtration. Its width was 50–100 m, the dip was northeast, at an angle of 75–80°. At a depth of 75–100 m, this zone was

distinguished fragmentarily. Beach subsidence and suffusion funnels formed along this zone in 2018.

2.  In the right bank junction on the resistivity maps at depths of 20 and 30 m, a linear anomaly of low resistivity 50–100 m wide was also distinguished. At a depth of 50 m and below, this zone became fragmented. A slight shift in the position of the anomaly in plan at different depths showed that the dip of the zone was close to vertical or southwestern at a steep angle (85°).

The resistivity of technogenic deposits at depths of 20–75 m varied from 9–18 Ohm·m to 45–100 Ohm·m. In the landfalls along the 605 m mark, the resistivity of the rocks reached 500–2000 Ohm·m, which is a sign of their frozen state.

In the places where dips formed in 2019 along the right bank, their direct connection with linear resistivity anomalies was not revealed. It was assumed that sinkholes were formed in places where ice-cement melts in the carbonate eluvium, as well as syngenetic massive ice in beach tailings. Well No. 1, drilled in the sinkhole, did not open a flooded fault, along which intensive water filtration and its discharge in the downstream were assumed. Therefore, the mechanism of water leakage is somewhat more complicated than previously thought. Probably, the thawed gravel eluvium played the role of drainage, through which water was filtered into the fault zones.

A three-dimensional geoelectric model made it possible to form a holistic view of the structure of bypass filtration zones in the right and left bank junctions. A diagram of the location of faults and filtration zones, as well as the choice of injection sites for plugging compositions, compiled on the basis of engineering geophysical studies, was presented.

**Author Contributions:** Conceptualization, K.T. and N.Y. (Nataliya Yurkevich); methodology, N.Y. (Nikolay Yurkevich), N.S. and V.O.; validation, K.T., N.Y. (Nataliya Yurkevich) and T.K.; formal analysis, N.Y. (Nataliya Yurkevich); investigation, K.T., N.Y. (Nikolay Yurkevich), T.K., N.S. and V.O.; resources, N.Y. and V.O.; data curation, K.T., N.Y. (Nikolay Yurkevich) and T.K.; writing—original draft preparation, N.Y. (Nikolay Yurkevich) and N.S.; writing—review and editing, K.T. and N.Y. (Nataliya Yurkevich); visualization, K.T., N.Y. (Nataliya Yurkevich) and V.O.; supervision, N.Y. (Nataliya Yurkevich); project administration, N.Y. (Nataliya Yurkevich); funding acquisition, N.Y. (Nataliya Yurkevich). All authors have read and agreed to the published version of the manuscript.

**Funding:** This research was funded by Ministry of Education and Science of Russian Federation, grant number FWZZ-2022-0029.

**Institutional Review Board Statement:** Not applicable.

**Informed Consent Statement:** Informed consent was obtained from all participants involved in the study.

**Data Availability Statement:** Data are available upon request to the corresponding author.

**Conflicts of Interest:** The authors declare no conflict of interest.

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
