# Peer review of "Engineering and Geophysical Research of the Tailing Dump under the Conditions of Growing Soils of the Base"

_applsci, doi:10.3390/app13074242_

Round 1

Reviewer 1 Report

Your manuscript needs to be revised. It requires a suitable methodology. The authors should write where the study area locates, how to collect the acquired data with what equipment, how many TEM stations are there, and how to validate the result. The references are not in a distinct style. Some referenced publications are in the Russian language, which requires English translation. The chapter discussion and conclusion still refer to some references, not describing the research results. This manuscript required major revision first before being reviewed.

Author Response

Thanks for your review. I tried to take into account all the comments in the new version of the article. Please see the attachment.
Best regards

Reviewer 2 Report

This Article uses Geophysical methods to evaluate  the ENVIRONMENTAL RISKS in Connection with the Technogenic Thawing of Permafrost SOILS in the Place Storage of Mining Enterprises in the Far North. The textual expression of the article makes it easy for readers to understand the main purpose of the article. The geophysical methods used in the article are appropriate, the charts are clear, the conclusions drawn are clear and reasonable, and the article has a good application prospect. It is recommended to publish.

Author Response

Thanks for your review. I tried to take into account all the rewiewer's comments in the new version of the manuscript.
Best regards

Reviewer 3 Report

Dear authors,

I think you should revise the current version of the study to make it more effective for readers who are concerned with this subject for possible publication in this journal.

 1.       Since the resolution of Figure 1, which shows a schematic representation of the tectonic faults, is very poor, it should be improved during revision. In the current figure, the left and bottom margins are invisible.

2.       I think you should mention the TEM method used in the research study in both the abstract and the introduction of the manuscript. Also, in the introduction, you should give some example citations that are similar to your research, as well as the main applications of the method. A sketch of the basic system of TEM may also be useful to a variety of readers of the journal in the methods section.

3.       It would be nice if you presented more technical details about the airborne TEM study. Why does the reader only learn this in the first paragraph of section 3.1? How many survey lines did you study? What is the altitude, flight time, instruments used, etc.?

4.       Please provide a detailed location map with survey line information. It would be very nice if you could show a picture from the survey

5.       Is it possible to display a 1D TEM anomaly with the inversion result? I am also interested in the inversion scheme used to interpret the collected data. Perhaps in addition to the resistivity distribution maps at depth, you can also present the results of the 1D inversion of the data obtained from the TEM study on the survey lines. Also, please explain how you obtained these maps from the 1D inversion.

6.       Figure 4 is not exactly legible, especially since it is difficult to see the explanations for 1-4 in this caption.

7.       Please use a unique explanation for the well number in the text and BH in the figures.

8.       Note whether Figure 4 refers to a depth of 75 m, which is explained in the first sentence of Section 3.1, or to 65 m, or vice versa. Even if you interpret the results with a depth of 150 m, we cannot see this in the manuscript.

9.       I am not sure your Figure 6 can really be interpreted as 3D if you created it using the results of the 1D inversion, which can give us changes in resistivity in the vertical direction. As I mentioned before, the study contains some gaps in terms of geophysical data acquisition, evaluation and interpretation, for which you do not provide detailed information that needs to be made up in the revision.

10.    You should turn your study into a scientific article instead of presenting the results of your research project, which for me really deserves attention.

 Best regards.    

Author Response

(The authors gave the same response as above.)

Reviewer 4 Report

Article

Engineering and geophysical research of the tailing dump under the conditions of growing soils of the base

The manuscript discusses an important task. But the conducted geophysical survey (TEM), must be supported with other geophysical technique as followed by the geophysical research as general.

Comments:

·         Abstract: need some editing.

·         The findings of the manuscript are not included in the abstract.

·         Keywords must not be exceeding 5-6 words.

·         Line (31): term “including” ???

·         Line (39): before 2000-2001, insert “years”.

·         Lines (46-49): the solution of the discussed problem was focused in environmental safety and energy loss. The manuscript must mention the tectonic elements that impacted the studied site.

·         Lines (58-61): the paragraph is not related with the previous paragraph.

·         Introduction: the manuscript must indicate the previous case studies, in addition to the literatures concerning the use of electrical resistivity for such cases.

·         Section “2.1”: object must change to “case study”.

·         Line (110): broken is not a scientific term.

·         Line (134): the manuscript mentions the cracks of 1-3 cm wide, which is not agreed with the figure (3).

·         Methods: the scientific background of the used geophysical technique (TEM) must be referenced.

·         Line (186): “The results of soundings by the other researchers”. What are the researchers?

·         The manuscript must be referenced well.

·         In the section “Method”: the manuscript must indicate the surveying parameters and the acquisition technique.

·         Figure 4: the legend ins not included.

·         Where the acquired geophysical surveys for different depths (20, 30, 50, 75, 100, 125, and 150m).

·         Figure 4 indicate Resistivity distribution map at a depth of 65 m, where the text not mention at this depth.

·        In the first part, the manuscript discussed the previous results conducted on the studied area. What are the differences between the results?

Author Response

(The authors gave the same response as above.)

Round 2

Reviewer 1 Report

The authors are asked to revise the manuscript in accordance with the general review suggestions and some notes from me (lines line 30; line 60-62 lacks of verb; lines 492, 497, 505 in references).

Author Response

Thanks for your review.

All comments of the reviewers were taken into account. The manuscript has been edited.

Reviewer 3 Report

Dear authors,

I congratulate you on your efforts to improve the manuscript during revision.

Best regards.

Author Response

(The authors gave the same response as above.)

Reviewer 4 Report

1. The abstract is still a qualitative description.

2. The abbreviation (PJSC) must be identified.

3. Paragraph (60-63) is not understandable.

4. The manuscript in need to the location map of the study area.

5. Lines (97-109): must be referenced.

6. Lines (109-137): must be referenced.

7. Figure 1: the legend is confused

8. Figure 1: the caption must be detailed and understandable.

9. Lines (147-152): must be re-edited in a scientific form ad referenced.

10. Figure 2: he caption must be detailed.

11. Figure 3: he caption must be detailed.

12. Lines (155-160): must be re-edited in a scientific form ad referenced.

13. Paragraphs (162-189) must e referenced. scientific background of the applied geophysical technique must be cited and referenced.

14.  Figure 4: the mentioned numbers must be described ad identified.

15. Figure 6: the mentioned numbers must be described.

16. The authors must prepare the 2D section for the performed borehole at the study area, to be compared with the finding of resistivity data.

17. Figure 7: some parts of the legend is not clear.

18. Discussion: need editing.

14. 

Author Response

(The authors gave the same response as above.)
